# *C. elegans* Apical Extracellular Matrices Shape Epithelia

**DOI:** 10.3390/jdb8040023

**Published:** 2020-10-06

**Authors:** Jennifer D. Cohen, Meera V. Sundaram

**Affiliations:** Department of Genetics, University of Pennsylvania Perelman School of Medicine 415 Curie Blvd, Philadelphia, PA 19104-6145, USA; jencohen@pennmedicine.upenn.edu

**Keywords:** *C. elegans*, apical extracellular matrix, cuticle, eggshell, glycocalyx

## Abstract

Apical extracellular matrices (aECMs) coat exposed surfaces of epithelia to shape developing tissues and protect them from environmental insults. Despite their widespread importance for human health, aECMs are poorly understood compared to basal and stromal ECMs. The nematode *Caenorhabditis elegans* contains a variety of distinct aECMs, some of which share many of the same types of components (lipids, lipoproteins, collagens, zona pellucida domain proteins, chondroitin glycosaminoglycans and proteoglycans) with mammalian aECMs. These aECMs include the eggshell, a glycocalyx-like pre-cuticle, both collagenous and chitin-based cuticles, and other understudied aECMs of internal epithelia. *C. elegans* allows rapid genetic manipulations and live imaging of fluorescently-tagged aECM components, and is therefore providing new insights into aECM structure, trafficking, assembly, and functions in tissue shaping.

## 1. Introduction

### 1.1. Apical Extracellular Matrix

Animals contact their environment via epithelial tissues, which create an impermeable barrier that separates the outside environment from the inside. The exposed outer surfaces of epithelia are coated with an apical extracellular matrix (aECM) that serves as a first line of defense against desiccation, pathogens, xenobiotics, and other environmental insults. Considerable evidence indicates that aECMs are also crucial for shaping epithelia, including biological tubes [1]. For example, aECMs such as the glycocalyx and surfactant maintain the narrow tubes of the mammalian blood vasculature and the lung [2,3].

Epithelia are polarized cells, with a basal surface facing the inside of the body and an apical surface facing the external environment or inside of biological tubes. Extracellular matrices that line the apical surfaces are distinct from those that line basal surfaces [4]. Basal surfaces are lined by basement membranes, extracellular matrices that form stiff, collagen-, laminin-and glycoprotein-rich sheets [4,5]. Apical surfaces are lined by diverse types of aECMs, which are laminar structures formed of lipid, sugar, and protein components [6,7,8]. By their repulsion of water, lipids protect against desiccation and hydrophilic small molecules [9,10]. Glycosaminoglycans, proteoglycans, and glycoproteins such as mucins may bind water molecules to swell and expand apical compartments, and may also prevent pathogens from accessing the apical membrane [8]. Fibril-forming proteins, such as collagens and Zona Pellucida (ZP) proteins, or carbohydrates such as chitin, can provide stiffness to shape apical surfaces and can assemble solid extracellular structures to form invertebrate cuticles [11,12,13,14].

Despite the importance of aECMs, the assembly mechanisms and compositions of aECM layers remain unclear. Functions of aECM are often inferred based on broad enzymatic removal or genetic depletion of entire glycosylation or lipid biosynthesis pathways. Thus, functions of individual matrix components are rarely described. Mechanisms by which aECMs affect cell shape also remain relatively obscure. For example, unlike basement membranes, which are known to impact the cytoskeleton via binding to integrins [5], it is not clear how most aECMs are anchored to the plasma membrane or whether they influence the cytoskeleton.

This lack of knowledge regarding aECM biology is due in part to challenges in visualizing aECMs, which are often translucent and invisible by standard light microscopy and are easily damaged by fixation protocols [15]. *Caenorhabditis elegans* is a genetically tractable model organism that allows rapid assessment of individual gene requirements and live imaging of fluorescently-tagged aECM components, and is therefore providing new insights into aECM structure, assembly and functions in tissue shaping.

### 1.2. Caenorhabditis elegans as a Model Organism for Studying Apical ECMs

The nematode *C. elegans* has several features that make it an attractive model system for studying animal aECMs. *C. elegans* contains aECMs with many of the same components (lipids, lipoproteins, collagens, ZP proteins, mucins, glycosaminoglycans, and proteoglycans) as mammalian aECMs, but it also has chitin-rich aECMs that may be more similar to those found in insects. *C. elegans* is optically transparent, enabling live imaging of these aECMs. Tubes of different sizes that nevertheless contain similar aECMs allow dissection of aECM function. Finally, powerful forward and reverse genetic approaches allow identification of key aECM components.

*C. elegans* generates many different types of aECMs in its various tissues and across its lifespan. Despite a maximum size of about 1 mm in length and only 959 somatic cells in the adult, *C. elegans* has a wide variety of epithelial cell types, each with their own aECMs. The *C. elegans* life cycle contains six stages: embryo stage (E), four larval stages (L1–L4), and the adult (A) stage. The L2 and L3 stages are different under non-crowded, ample-food conditions, which are optimal for reproductive growth, and under non-ideal conditions such as crowding, absent food and high temperatures, when the animal forms stress-resistant dauer larvae (Figure 1) [16]. A flexible, collagenous cuticle lines most adult and larval external epithelia [17]. Between larval stages and between the L4 and adult are the molts, during which the animal synthesizes a new cuticle and sheds its old cuticle [18]. Embryos and molting larvae synthesize a transient glycoprotein-rich “pre-cuticle” between the old and new cuticles [19,20,21,22]. In addition to the cuticle and pre-cuticle, *C. elegans* has a number of other aECM types. For example, a chitinous eggshell surrounds the developing embryo [23,24,25], and the *C. elegans* pharynx contains a rigid chitinous cuticle with “teeth” that pulverize bacteria for digestion [25,26,27]. Internal epithelia such as the gut and uterus also have poorly described but intriguing non-cuticular aECMs. Below, we summarize the structures and functions of these various different aECMs.

## 2. The Eggshell

*C. elegans* embryogenesis occurs within an eggshell that both protects and shapes the developing organism. Many aspects of eggshell structure and assembly have been reviewed recently [24]. The eggshell is produced primarily by the zygote, starting immediately after fertilization of the oocyte in the spermatheca, with some contribution from the uterus [29]. The final eggshell contains five morphologically and biochemically distinct layers composed of proteins, chitin, chondroitin proteoglycans (CPGs), and lipids, plus a peri-embryonic layer that apposes the embryo [23,25,30,31] (Figure 2). Besides the peri-embryonic layer, whose relationship with the later-forming embryonic sheath is unclear, these eggshell layers are separate from the pre-cuticle matrix layers that form later around the embryo (see below).

### 2.1. Building the Eggshell

The eggshell is constructed by sequential waves of matrix secretion and deposition, with the outermost layers added first, followed by more internal layers [31]. Prior to ovulation, the vitelline layer is secreted. Upon fertilization, the vitelline layer is remodeled and a chitin-rich layer is produced beneath it [25,32]. After fertilization, meiotic progression triggers further temporally distinct steps of eggshell secretion, including delivery of cortical granules to form the chondroitin proteoglycan (CPG) layer at anaphase I and delivery of lipids to form the permeability barrier layer at anaphase II [30,31]. Finally, uterine-derived proteins can attach to the outer eggshell surface, though it is not clear if any actually incorporate into its matrix [29].

The permeability barrier layer of the eggshell is built from fatty acids imported into the developing oocyte from the mother’s somatic tissues [24,31,33,34]. Loss of this permeability layer allows entry of water and small molecules through the eggshell and perturbs development of the peri-embryonic layer and the embryo [32,34]. Supplementing the animals’ diet with specific poly-unsaturated fatty acids (PUFAs) rescues embryo viability and defects in the permeability barrier and peri-embryonic layer in mutants with defects in fatty acid transport [35], indicating that PUFAs may be the main components of this layer.

A few protein components of different eggshell layers have been identified. The secreted chitin binding domain protein CBD-1 and the secreted mucins PERM-2 and PERM-4 form a complex within the outermost vitelline layer [30]. CPG-1 and CPG-2 are nematode-specific chondroitin proteoglycans within the CPG layer [31]. The transmembrane extracellular leucine-rich-repeat only (eLRRon) protein EGG-6 is a potential peri-embryonic layer component or interactor based on its mutant phenotype and its presence within the embryonic pre-cuticle [21], but its specific localization with respect to the eggshell has not yet been determined. The shared requirement for EGG-6 for both eggshell and pre-cuticle organization raises the possibility that epidermal pre-cuticle could arise in part from an earlier eggshell layer. Although ZP proteins are major components of the mammalian eggcoat [36], none have been described in the *C. elegans* eggshell.

Defects in one eggshell layer can lead to defects in assembly of subsequent layers, but not always in a predictable manner. For example, loss of the vitelline layer component CBD-1 also disrupts formation of the internal layers [30,31]. Similarly, loss of chitin leads to defects in the permeability layer [32] and loss of enzymes that produce fatty acids required for the permeability layer causes defects in the peri-embryonic layer [31,32,33,34]. On the other hand, loss of PERM-2 or PERM-4 mildly disrupts the vitelline layer and the permeability layer without greatly disturbing the overall integrity of the intervening chitin or chondroitin layers [30]. The data support a hierarchical assembly model whereby early deposited outer layers help constrain subsequently secreted factors to form inner layers, but also suggest possible communication between distantly placed layers. It is not yet clear if inner layers also could aid in remodeling or maintenance of previously secreted outer layers.

### 2.2. The Eggshell Shapes Early Development

The eggshell influences early stages of *C. elegans* embryogenesis. Before egg laying, the early-deposited chitin and vitelline layers stiffen the eggshell [37] and protect the new zygote from fragmenting as it transitions from the spermatheca to the uterus [32]. Mutation of key eggshell components or chemical removal of eggshell outer layers during the meiotic divisions leads to a number of severe defects in polar body extrusion, centrosome movement to the cortex, proper cell division axes, and embryo elongation [32,38,39,40,41,42]. Often, actin is also mislocalized, and this may drive the observed phenotypes [32,39,40], although the relationship between actin and the eggshell is not known. Chemical removal of the vitelline and chitin eggshell layers after the meiotic divisions can produce abnormal cell division polarities, likely due in part to disrupted cell–cell interactions [43,44,45]. In contrast, enzymatic removal of only the chitinous layer results in mild defects in embryonic elongation [46]. The eggshell likely offers a combination of mechanical structure and signaling to shape the embryo over time.

## 3. Pre-Cuticular aECMs

Pre-cuticular aECMs form just prior to the 1.5-fold stage of embryogenesis (Figure 3A–C), disappear prior to hatching and are replaced by cuticle, and then reappear before each larval molt [20]. A transient pre-cuticle can be detected in every tissue that later secretes collagen-based cuticle, including the epidermis and various interfacial tubes that connect the epidermis and the external environment to internal tissues.

Pre-cuticle typically contains glycoproteins of the ZP, eLRRon, and lipocalin families [20,21,47,50,52,53]. Some of these proteins reappear at each molt, and are present in many tissues, while others may be present in only a subset of tissues or stages. For example, the ZP protein NOAH-1 is strongly present in the epidermal sheath pre-cuticle but not visible in interfacial tubes of the embryo, whereas a different ZP protein, LET-653, has the converse pattern [47,52] (Figure 3B). Thus, there is no single type of pre-cuticle, but rather a set of related pre-cuticles, each specialized for a particular epithelial tissue.

Pre-cuticle aECMs are present during the major periods of epithelial morphogenesis, and play important roles in tissue shaping and in patterning the cuticles that eventually replace them. Loss of pre-cuticle components results in severe defects in body shaping, tube shaping, and cuticle structure.

### 3.1. The Epidermal Sheath Elongates the Embryo

The embryonic epidermal sheath overlays the epidermis and promotes embryo elongation (Figure 3A). Although the sheath was first visualized by scanning electron microscopy several decades ago [22], the first embryonic sheath components were identified more recently. These include the ZP proteins NOAH-1, NOAH-2, and FBN-1, the eLRRon proteins SYM-1, EGG-6, and LET-4, and the lipocalin LPR-3 [21,47,50,52,53]. Additional factors are seen specifically over seam cells that produce alae ridges (see below) or in interfacial tubes (LET-653, DEX-1) [20,52].

The *C. elegans* embryo elongates into a worm shape via a combination of actinomyosin-based contraction and pre-cuticle stabilization [54]. During the earliest phases of embryo elongation, actin-myosin filaments in the seam epidermis constrict circumferentially, and actin and the sheath aECM appear to reorganize together to support the new body shape [22,47,54]. Once body muscle contractions begin at the two-fold stage, these exert additional tension on the newly developed hemidesmosomes that cross the epidermis to link the sheath with underlying muscle. This tension triggers an actin severing mechanism that further shortens circumferential actin filaments throughout the epidermis [54]. In the absence of key sheath components NOAH-1 or NOAH-2, the embryo begins to elongate but then retracts and sometimes ruptures. In the absence of multiple sheath components, the remaining aECM detaches from the epidermis and muscle and even less initial elongation occurs [22,47]. Therefore, the sheath aECM is not only needed to stabilize shape changes induced by cytoskeletal forces, but also to allow those changes in the first place. Whether the aECM also generates some of the constriction force remains to be investigated.

Despite the apparent connection between the sheath aECM and hemidesmosomes, it remains unclear how they are connected across the plasma membrane. MUP-4 and MUA-3, which connect the cuticle to hemidesmosomes, are possible candidates [47,55,56,57]; however, their single mutant phenotypes arise later than those of sheath mutants. NOAH-1, NOAH-2 and FBN-1 do have transmembrane domains, but most ZP proteins are cleaved away from their transmembrane domains as a pre-requisite for subsequent polymerization [58]. SYM-1 and LPR-3 do not have transmembrane domains [53,59], and transgenic experiments suggested that the LET-4 transmembrane domain is not essential for its tissue-shaping functions [21]. Therefore, the relevant transmembrane linkers remain to be identified.

### 3.2. A Luminal Precuticle Shapes the Narrow Excretory Duct and Pore Lumens

The excretory system is an osmoregulatory organ that contains three tandem, single-celled tubes: the canal, duct, and pore cells (Figure 3D) [60,61]. The canal extends four lumenized arms along the animal’s body cavity, from which it presumably exchanges osmolytes. The canal attaches to the duct cell and the duct attaches to the pore, which releases excretory contents into the outside environment.

During embryogenesis, as the tubes of the excretory system elongate, the duct and pore are lined by a set of pre-cuticular aECM components, including the lipocalin LPR-3 [53], the ZP protein LET-653 [20,48], the nidogen-domain protein DEX-1 [52], and the eLRRon proteins LET-4 and EGG-6 [21]. In addition, the lipocalin LPR-1 appears to affect the function of this aECM, but does not stably incorporate into it [62,63]. Loss of any one of these components causes lumen collapse and dilation in the duct and pore tubes, leading to fluid retention and rod-like lethality at the first larval stage; many pre-cuticle components were initially identified based on this phenotype [20,21,52,53,62,63].

How pre-cuticle shapes the duct and pore tubes remains unclear. Ectopic expression of one pre-cuticle component, LET-653, was sufficient to expand the gut lumen [20], suggesting an intrinsic, lumen-expanding activity (Figure 3E). Nevertheless, in *let-653* mutants, duct and pore luminal defects typically appear after the time that LET-653 protein would normally have been cleared, suggesting improper assembly of later pre-cuticle or cuticle components that more directly impact lumen structure [20]. Although LET-653 re-appears within the duct during each molt cycle, its lumen shaping activity is only required in the embryo, suggesting that the pre-cuticle counters forces specifically present during morphogenesis (Figure 3E) [20]. It is possible that pre-cuticle components distribute actin-myosin-dependent contractile forces similar to the role proposed above for the epidermal sheath [54], and/or that they create a luminal scaffold around which the narrow duct lumen can elongate.

### 3.3. Pre-Cuticular and Sensory aECMs Anchor the Pharynx and Sensory Organs to the Epidermis

The pre-cuticle and other sensory matrix factors have a critical role in shaping the embryo’s developing nervous system and buccal (mouth) cavity. The buccal cavity, at the anterior end of the pharynx, is surrounded by bundles of sensory neurites and glia forming a rosette (Figure 3C,C’) [64,65,66]. During embryonic elongation, several aECM factors help these organs remain anchored to the anterior epidermis while the pharynx elongates and the neuronal and glial cell bodies actively migrate towards the posterior [49,50] (Figure 3C’).

Three matrix proteins have been reported to shape the buccal cavity. FBN-1 is a large fibrillin-like ZP protein present within the epidermal sheath and also secreted into the buccal cavity [50,52]. DYF-7 is a neuronally-expressed ZP protein, and DEX-1 is an epithelial and glial-expressed Nidogen-and EGF-domain protein [49,52]. Loss of any of these factors can cause the buccal cavity to over-elongate and the pharynx to ingress within the worm’s body during embryonic elongation [50,52,67]. FRET sensors demonstrated that the elongating pharynx exerts an inward pulling force on the anterior epidermis and buccal cavity [50]. The various ECM factors appear to resist pulling forces from morphogenesis to maintain the shape and integrity of the entire nose region (Figure 3C’).

Similarly, DYF-7 and DEX-1 anchor neurites and glia to the epidermis [49,51]. Sensory neurons in the nose tip (amphid) and tail (phasmid) extend neurites into the environment through two sets of wrapping glia, the sheath and the socket, both of which have epithelial tube-like characteristics [51]. The socket glia are situated at the external body surface and are coated in a pre-cuticle or cuticle (Figure 3C’). In contrast, the neighboring sheath glia are lined by a non-cuticular aECM [51,66]. Transmission electron microscopy of embryos reveals a fibrillar matrix within the tube-like lumens of the sheath and socket glia [51]. The ZP protein DYF-7 is present in this matrix as observed by fluorescence microscopy, and in *dyf-7* mutants, much of the fibrillar structure of this matrix is absent. DYF-7 is therefore a presumed fibrillar component of this sensory-specific matrix. DYF-7 is localized to the dendrite tip by *par-6* [64] and the ciliary transition zone genes *ccep-1* and *nphp-4* [68], suggesting that multiple intercellular systems converge to attach this sensory aECM to glial cells, neurons, and to the epidermal aECM. However, the precise connections between the various cells and matrices remain to be elucidated.

The diameter of sheath and socket glia lumens also are determined in part by DYF-7 [51] and by the coordinated action of the Patched-related genes *daf-6* and *che-14,* which were proposed to regulate secretion or endocytosis of aECM factors [69,70,71,72,73]. *daf-6* mutants have closed sockets and expanded sheath lumens [71], while *che-14* mutants accumulate vesicles in the amphid lumen [69]. In addition, *daf-6* mutants also have *dyf-7*-like dendrite anchoring defects [74]. DAF-6 is localized to the glial apical membrane by the apically secreted PLAC-homology domain protein DYF-4, whose loss phenocopies *daf-6* mutations [74].

### 3.4. The Vulva Lumen Is Shaped by a Multi-Layered Pre-Cuticular aECM

The vulva is a relatively large tube that connects the uterus to the outside environment to allow for egg laying [75]. It is comprised of twenty-two cells of seven different types derived from either Ras-dependent (primary) or Notch-dependent (secondary) vulva precursor cells [76] (Figure 3F). During L4 stage, the vulva first expands dramatically from a simple invagination to a large lumen via the action of chondroitin glycosaminoglycans and actin-myosin constriction [77,78,79,80,81,82]. Next, the vulva tube is shaped into a narrow, slit-like channel via further cell shape changes, rearrangements, and the action of its pre-cuticular aECM (Figure 3G) [48,83]. This multicellular tube is large enough to allow for the visualization, and thus the dissection, of its pre-cuticular aECM’s spatial, temporal, and functional organization [48].

Chondroitin glycosaminoglycans (GAGs) are crucial for initial vulva lumen expansion (Figure 3G). Electron microscopy reveals that the inflated vulva lumen is entirely filled with a granular matrix that likely corresponds to these GAGs and CPGs [48]. Loss of chondroitin GAGs or GAG sulfation causes dramatically narrowed or “squashed” vulvas (Sqv phenotype) [79,80,81,84,85]. The current model holds that chondroitin absorbs water molecules to expand the vulva lumen like a sponge. However, in addition, chondroitin appears to work with pre-cuticle components to constrain and shape the lumen more locally [48].

GAGs are typically attached to protein carriers to form CPGs, but the relevant carriers for vulva expansion are not yet known. There may be multiple redundant carriers since none were identified in the original genetic screens for *sqv* mutants [84]. Mass spectrometry studies have identified twenty-four *C. elegans* proteins that have chondroitin GAG attachments [23,86], and at least one pre-cuticle component, FBN-1, is among these, but so far none of the corresponding mutants have been described to have vulva expansion defects [23,48]. Instead, *fbn-1* mutants have defects in later stages of vulva eversion [48]. Future studies of double mutants may be needed to determine which CPGs work together to inflate the vulva lumen.

Most of the pre-cuticle proteins identified to date are found in the developing vulva, and these proteins each have distinct localization patterns that mark different aECM layers lining specific vulva cell types [20,48,53]. For example, the ZP protein LET-653 and the lipocalin LPR-3 label slightly offset membrane-proximal pre-cuticle layers [53], whereas LET-653 also labels a stalk-like core structure in the central part of the lumen (Figure 3F’) [20,48]. Furthermore, different pre-cuticle factors label the surfaces of primary-or secondary-derived cell types at different stages (Figure 3F,F’). Vulva aECM contents change dramatically over short timeframes during tube morphogenesis, before eventually being replaced by cuticle. These reproducible spatial and temporal patterns suggest highly regulated mechanisms for pre-cuticle assembly and disassembly, and the vulva is an ideal organ system for dissecting these mechanisms.

Despite the elaborate structures decorated by pre-cuticle proteins, only mild defects in the vulva lumen shape have been detected in single mutants [20,48]. *let-653* loss had more dramatic effects when combined with chondroitin perturbations, again suggesting redundant contributions of multiple aECM factors to lumen shape [48].

### 3.5. The Pre-Cuticle Patterns the Cuticle

Towards the end of morphogenesis, once tissues have taken their shapes, pre-cuticle components disappear and are replaced by cuticle. How this transition occurs is not understood, but it is likely to be gradual rather than abrupt, with the pre-cuticle serving as a scaffold or template to which various collagens and other cuticle components are added. TEM imaging of discrete timepoints in the embryo suggested sequential addition of inner cuticle layers without obvious loss of outer layers [19], much as described for the eggshell [31]. Different cuticle collagens become expressed at different times during the molt cycle [87,88,89], so some could be present within both the pre-cuticle and the cuticle. When imaged directly, different pre-cuticle components disappear at different time points [48], suggesting gradual dismantling of the initial pre-cuticle structure. Consistent with a role in patterning the cuticle, many pre-cuticle mutants have compromised cuticle barrier functions, abnormal levels or exposure of cuticle surface lipids, and/or structural defects in cuticle alae ridges in larvae or adults [20,21,52,53].

## 4. Collagen-Based Cuticles and the Molt Cycle

The cuticles that line external epithelia of *C. elegans* larvae and adults are multi-layered structures composed of many different collagens and lipids, as well as other poorly characterized glycoproteins and insoluble proteins collectively termed cuticulins (Figure 4) [17,90,91,92]. Cuticles attach to muscles to allow locomotion, provide a barrier to protect the organism from rupture, desiccation, and pathogens, and also shape (or maintain the shape of) tissues [93,94,95,96,97,98,99,100]. Between each larval stage, *C. elegans* molts into a new cuticle that is unique in structure and collagen composition for that life stage, but how these cuticles differ functionally is not clear [101,102]. Molting occurs four times, and then the adult cuticle remains present throughout the rest of the organism’s life (Figure 1) [18]. The expression of many genes rises and falls in accordance with distinct phases of the molt cycles [87,88,103,104], controlled by a molecular clock related to the circadian molecular clock of other organisms [105]. The molt process was recently extensively reviewed [18].

### 4.1. Epidermal Cuticle Structure and Function

The epidermal cuticles have been the most intensively studied of all *C. elegans* aECMs [17,109]. A few dozen collagens and a smaller set of non-collagen protein components of this cuticle have been identified [110,111,112]. Most of the collagens are related to the mammalian FACIT (Fibril-Associated Collagens with Interrupted Triple helices) family, although some have unusual features not seen in mammalian collagens [110]. The *C. elegans* genome encodes > 170 members of this family, many of which presumably contribute to the cuticle (or pre-cuticle). The non-collagen cuticle components include several ZP domain proteins, lectins, and other unknown glycoproteins [90,91,93,113]. Finally, biochemical studies suggest that a variety of lipids are also present, including free fatty acids, phospholipids, triglycerides, and other more complex lipid types [107].

Cuticle components are organized into discrete layers (Figure 4A). Enzymatic digestion studies suggest that, in adults, collagens predominate in the basal-most striated layers, while cuticulins and other glycoproteins are present in the cortical and surface layers [102]. Glycosyltransferase (Bus) and nucleoside transporter (Srf) genes generate the glycoproteins of the outermost layer (Figure 4A) [93,95,96,97]. In adults, the basal and cortical zones are separated by an intermediate layer, which appears fluid-filled but contains connecting collagenous struts (Figure 4A). Mutants in the collagens BLI-1, 2, and 6 lack struts and have a Blistered (Bli) phenotype in which the cortical and basal layers of the cuticle detach from one another [102,114,115]. Beyond general categories, the specific contents of each cuticle layer are still little known.

One important function of the cuticle is to serve as a barrier against the penetration of toxins and other molecules. This barrier function is likely conferred by a lipid-rich layer that can be visualized by DiO/DiI staining or TEM (Figure 4A) [116,117]. Loss of enzymes that produce long-chain fatty acids (*pod-1*, *fasn-1*, *acs-20*) result in cuticle barrier defects, thinning of this lipid layer, and alae defects [94,118]. It remains unclear how lipids are secreted and incorporated into the cuticle, or if they are present in more than one layer. Interestingly, the transcription factor CEH-60 seems to act in the gut to affect cuticle barrier formation, suggesting that cuticle lipids may be derived from multiple tissue types [119].

The cuticle connects to the underlying epidermis via the matrilin-and fibrillin-related transmembrane proteins MUP-4 and MUA-3 [55,56,57]. It is not known which specific cuticle components are involved in binding these linkers, but annular furrow collagens (see below) are good candidates. The connection occurs at apical hemidesmosomes (Figure 4A), which contain the plectin VAB-10a and serve as attachment sites for cortical actin bundles, microtubules, and intermediate filaments [99,120]. Apical hemidesmosomes are in turn linked to basal hemidesmosomes via intermediate filaments, which span the epidermis [121]. Basal hemidesmosomes connect to the underlying body muscle via the transmembrane protein LET-805/myotactin [99,122]. Thus, the cuticle is anchored to both the epidermis and body muscle via the cytoskeleton. These attachments must be remodeled at each larval molt, but how this occurs and whether temporary attachments are formed is not known.

### 4.2. Alae and Annuli

Notable morphological features of epidermal cuticle are its two sets of cuticle ridges: annuli and alae (Figure 4B). Different cuticle components are responsible for building each of these features.

Annuli are circumferential ridges that are present at all stages (Figure 4B). The collagens DPY-2,3,7,8,10 are required to form annuli; of these, only DPY-7 and DPY-10 have been visualized, and both localize only at the low points or “furrows” of annuli, where the cuticle attaches to hemidesmosomes [123]. In contrast, the collagen DPY-13 is present on the raised portions of annuli [123]. Disruption of annuli triggers upregulated autophagy and hyperosmotic, detoxification, and innate immune responses in many tissues [124,125,126], indicating that one function of annuli is to sense and transmit information about cuticle damage.

Alae are longitudinal cuticle ridges that form above the lateral epidermis or “seam” cells (Figure 4B). They are present only in first stage (L1) larvae, dauer larvae, and adults and have distinct appearances at these different stages [17,101]. Several ZP proteins (CUT-1/3/5/6) localize to the alae of one or more stages, and are required to generate or shape them [14,90,91,92,113,127,128,129]. For example, CUT-1 promotes formation of dauer alae, CUT-3 promotes formation of L1 alae, and CUT-4 promotes formation of adult alae [14]. Both CUT-5 and the nidogen domain protein DEX-1 promote formation of alae in L1s and dauers, but not adults [14,130]. The collagens DPY-2, DPY-3, DPY-10 [123], DPY-5, DPY-11 and DPY-13 [124], and the secreted proline-rich-repeat protein MLT-10 [131] are all required for normal adult alae morphology; it’s not clear whether they are also required for development of L1 or dauer alae. Pre-cuticle components are also important for alae shaping [52,53,108,130]. Alae are thought to form by circumferential constriction of the seam epidermis and/or polymerization of membrane-proximal ZP matrix layers in cuticle or pre-cuticle, which leads to buckling of the overlying cuticle layers (Figure 4C) [14,108]. The purpose of alae is not known, but two possible functions are to aid in locomotion or to serve as a reservoir of extra cuticle material to accommodate animal growth.

### 4.3. Epidermal Cuticles Maintain Body Length and Girth

Sidney Brenner’s original genetic screens in *C. elegans* identified many body shape mutants that turned out to encode cuticle collagens [17,132,133]. The majority of these collagen mutants are notably short and fat (Dumpy; Dpy), while a smaller set are excessively long and thin (Long; Lon), develop cuticle delaminations (Blistered; Bli) or have twisted body axes (roller; Rol). Therefore, cuticle collagens can affect body shape in multiple ways.

The elongated worm shape of *C. elegans* arises through actomyosin-based constriction of the epidermis during embryogenesis with contributions from the pre-cuticular epidermal sheath (see above) [3,22,47,54,98,100]. Early studies of *sqt-3* cuticle collagen mutants (which are Dpy) revealed that this collagen was not needed for body elongation per se, but rather for maintenance of the elongated state [22]. This fits with the idea that most cuticle secretion happens after morphogenesis, and that tissue anchorage to the cuticle stabilizes the current shape established by the pre-cuticle.

Many matrix mutants that lack alae have shorter and wider seam cells and a shorter and wider body shape [14,123,124,130], suggesting that aECM factors that generate the alae also play a role in seam cell and body constriction.

The *C. elegans* TGFβ signaling pathway affects body size, at least in part by regulating collagen gene expression. Mutants with reduced TGFβ signaling have a small (Sma) body size despite normal cell numbers, while mutants with increased TGFβ signaling are Lon [134,135,136,137,138]. These signaling mutants also have changes in intracellular lipid storage and in cuticle surface lipid accumulation [116,139,140]. Direct or indirect targets of TGFβ -regulated Smad transcription factors include the collagen genes *rol-6*, *col-41*, *col-141*, and *col-142*, whose loss and/or overexpression also impacts body size [141]. Interestingly, mutations in some collagens also reduce expression levels of the TGFβ ligand DBL-1, suggesting a positive feedback loop whereby cuticle structure maintains proper TGFβ signaling [142]. Relationships between TGFβ signaling and extracellular matrix organization also have been found in mammalian systems [143], and *C. elegans* could be a good system for understanding some of these connections.

### 4.4. Cuticles of Interfacial Tubes

Collagenous cuticles also line the interfacial tubes that connect the epidermis and the external environment to internal tissues (Figure 3F), but in most cases little is known regarding their specific composition. Since these tubes have somewhat different pre-cuticle components compared to the epidermis (Figure 3B,C, see above), it is possible that they also have distinct cuticle collagens or other components. These cuticles do have some unique morphological features. For example, electron micrographs show dramatic vertical striations within the cuticle of the buccal cavity [144] and orbital ridges surrounding the opening of the excretory pore [28]. The adult rectum is uniquely susceptible to bacterial adhesion and infection [145,146]; whether this is due to a unique cuticle makeup remains to be determined.

The most intensively studied interfacial tube cuticle has been that of the male rays. Rays are cuticle-lined glial tubules through which sensory neurites extend [147,148]. Cuticle proteins important for building the male rays include the ZP domain protein RAM-5 [149], the short, secreted peptide LON-8 [150], and the collagens RAM-2, COL-34, and SQT-1 [149,151]. Several proteins implicated in collagen processing are also required for male ray morphology, including the ADAMTS metalloprotease, ADT-1 [152], the collagen-modifying thioreductase DPY-11 [153,154], and the prolyl hydroxylase DPY-18 [151]. Tunicamycin treatment, which prevents glycosylation, results in severely deformed male rays with cuticular defects, indicating that glycoproteins are also directly or indirectly important for building the ray cuticle [155].

### 4.5. Regulation of the Molt Process

Molting involves the coordinated assembly and disassembly of pre-cuticles and cuticles. Many gene products are required for molting [156], including proteases [157,158,159], pre-cuticle [53] and cuticle components [131], endocytic proteins [160], metabolic enzymes [103,118,161], secretory pathway genes [69,162], and Hedgehog-and Patched-related genes [69,163]. The expression of many genes rises and falls in accordance with distinct phases of the molt cycle [87,88,103,104]. This process was recently extensively reviewed [18].

To secrete large amounts of cuticle proteins during molt, *C. elegans* coordinately alters secretory and stress pathways. Lysosome-related organelle (LRO) morphology changes dramatically in epidermal cells before and during ecdysis, indicating that LROs may be particularly important for molting [162]. In addition to their roles in protein and lipid degradation, these acidic vesicles are central hubs of the secretory pathway, that are able to send and receive cargo from endocytic, secretory, or lysosomal vesicles [164]. The high level of secretion presumably required to build a cuticle also relies on stress pathway regulation. Upregulation of ER stress proteins occurs in a developmentally regulated fashion during molting [165].

### 4.6. The Cuticle Changes during Aging

Degradation of the cuticle may be one major cause of aging-induced health decline. The adult *C. elegans* cuticle must remain intact throughout the organism’s two weeks of adulthood, even as *C. elegans* continues to grow in both length and width. Over the course of one week of adulthood, cuticle structure becomes irregular and its stiffness declines [166]. Some cuticle collagens gradually decrease in expression level throughout adulthood, and excess expression of specific cuticle collagens throughout development can prolong life [167]. Food deprivation, pathogen infection, or other stressors also can trigger protective upregulation of collagen gene expression [167,168,169,170]. These observations suggest that cuticle degradation contributes to the aging process, but that the adult cuticle can be repaired when newly secreted aECM components are provided.

## 5. Chitin-Based Pharyngeal Cuticle

The pharynx, or foregut, is a myoepithelium with a cuticle that is different than the cuticle of the rest of the body. The pharyngeal cuticle contains the carbohydrate chitin [25], and may not contain collagen. It appears to contain different types of secreted matrix proteins, including pharynx-specific mucin-like proteins [171] and many proteins predicted to form amyloid (encoded by the *abu*/*pqn* paralog group genes) [87]. Consistent with the presence of amyloid, the pharyngeal cuticle stains with Congo Red, a marker for amyloid [87].

The pharynx transports bacteria through its lumen into its posterior end called the terminal bulb (TB). The TB contains teeth-like cuticular specializations in its grinder that break bacteria before passing them into the gut [26,27] (Figure 5A,B). The cuticle that lines the pharyngeal lumen and makes up the grinder is key to *C. elegans* feeding [172,173,174]. Loss of either chitin or the predicted amyloid-forming protein ABU-14 causes lumen shaping and/or grinder defects which results in poor transport and mashing of bacteria [25,87].

Like the body cuticle, the pharyngeal cuticle is replaced with each larval molt cycle. However, in contrast to the body cuticle, which grows continuously, the pharyngeal cuticle grows only during the molt [87,177], perhaps due to the higher rigidity of this aECM. Following enzymatic digestion of the old cuticle, part of the old cuticle is expelled through the mouth, while the rest is swallowed [87,175]. There appears to be a pharyngeal pre-cuticle in the embryo [52], but its contents are largely uncharacterized, and it’s not clear if it reappears during molt. Dedicated proteases, such as CPZ-1 [157], NAS-6, and NAS-7 [175,178], are required for pharyngeal cuticle removal during the molt.

Ultrastructural studies revealed that the pharyngeal grinder has five distinct layers that assemble in a sequential manner during molts (Figure 5B) [175]. It is not yet clear which molecular components are present in each layer. Interestingly, during grinder synthesis, pharyngeal muscle cells transiently lose their striated muscle-like appearance and take on a more epithelial and secretory, vesicle-filled, appearance, suggesting a toggling between the two aspects of their cell identity in order to build cuticle [175].

The pharyngeal cuticle has evolved specialized features in different nematodes. In the facultative predator nematode *Pristionchus pacificus*, the cuticle at the transition between the buccal cavity and the pharynx can take on two forms [179]. The first, stenostomatous, contains a single chitinous tooth, and is sufficient for ingesting bacteria. The second, predatory morph, eurystomatous, includes two teeth that can pierce the cuticles of other nematodes. The choice between these mouth-forms is made based on environmental inputs, including pheromones, diet, and habitat [179,180,181,182]. These factors converge on the neuronally-expressed sulfatase EUD-1 and the α-*N*-acetylglucosaminidases NAG-1 and NAG-2, which then activate chromatin modifiers to promote either the predatory morph or the bactericidal morph, respectively [183,184,185]. The *C. elegans* genome encodes several orthologs of *eud-1, nag-1*, and *nag-2* [186], but it is not known whether they impact cuticle or pharyngeal aECM structure.

## 6. aECMs of Internal Epithelia

Relatively little is known about the composition or functions of the non-cuticular aECMs that line *C. elegans* internal epithelia, such as those of the gut or uterus. However, these tissues do contain aECMs that likely play important roles in tissue shaping and/or function.

### 6.1. The Gut

The *C. elegans* gut is composed of sixteen ciliated cells lining a lumen [187]. Transmission electron microscopy (TEM) reveals that cilia are bathed in a ~1 micron thick electron-dense aECM from which bacteria appear to be excluded (Figure 5C) [176,188,189]. This membrane-proximal aECM layer resembles the mucin-rich glycocalyx of the mammalian gut [8]. Several secreted proteins can be detected within the larval gut lumen, including the lectins LEC-6 and LEC-10 [190], the leucine aminopeptidase LAP-1 [191], and the bacteria-killing lysozyme ILYS-3 [192]. The gene *f57f4.4,* which encodes a large secreted protein, is also expressed in the gut [193]. Future research is needed to determine whether these proteins contribute to the gut aECM, and to determine their roles in gut function.

### 6.2. The Uterus

The *C. elegans* uterus is a large, multicellular lumen into which fertilized eggs are deposited before passing through the vulva during egg laying. The uterus expands dramatically during L4 stage [194] and fills with an amorphous aECM visible by TEM (Figure 6). A set of secreted proteins and lipids are present in the uterus throughout adulthood (VIT-6, ULE-1–5) [29,195]. One of these, ULE-5, is deposited onto the surface of the eggshell, while the rest are retained within the uterine lumen and surround developing embryos [29]. Functions for these proteins are not described, and it is not clear whether these or other proteins are incorporated into the uterus aECM.

### 6.3. The Excretory Canal Cell

The excretory canal cell extends four long lumenized tubules along the length of the worm, through which it is presumed to exchange osmolytes with the body cavity. It then drains its contents through the excretory duct and pore (Figure 3D) [61]. Although some electron micrographs show a meshwork within the canal lumens [21,28], the contents of this aECM are not known. At least one ZP protein, DYF-7, is expressed in the canal cell and may contribute to its aECM [49,52].

## 7. Outstanding Questions Regarding aECM

There remain many unanswered questions about how aECMs assemble, connect to underlying cells, and shape epithelia. aECMs are challenging to study as they generally do not develop fully in cell culture and can be destroyed by the fixation required to visualize aECMs in many animal systems. *C. elegans* is an excellent model for addressing how aECMs assemble and function, as it offers a set of aECMs that can be visualized without fixation in vivo.

### 7.1. How Are aECM Components Trafficked to Apical Cell Surfaces?

Building an organized aECM requires that many different components are trafficked to the apical surface, and that matrix assembly occurs only once these components have arrived in their proper locations. We know little about the vesicular compartments through which most aECM proteins or lipids travel, or when or where these components become exposed to relevant conditions and partners for assembly into gels or fibrils. Studies of collagen-, mucin-or lipid-rich matrices in mammals have identified some specific vesicular compartments important for matrix delivery, but these compartments are not well understood, and whether *C. elegans* uses similar compartments is not yet known.

Collagens and other extracellular matrix cargos are thought to require extra-large vesicular or tubular compartments in order to traffic through the secretory system [196]. In *C. elegans*, as in other organisms, coat complex II (COPII) appears to be required for collagen secretion [197]. Recent studies in mammals have identified the transport and Golgi organization (TANGO1) protein as important for ER-to-Golgi trafficking of large proteins [198], but no TANGO1 ortholog has been identified in *C. elegans*. Instead, efficient secretion of at least some cuticle collagens requires the evolutionarily conserved ER protein TMEM131, which binds to the TRAPPC8 component of the Transfer Particle Protein III (TRAPIII) COPII-tethering complex [199].

In mammalian goblet cells of the lung and gut, large acidic vesicles deliver highly condensed mucin packets that expand once exposed to the higher pH of the extracellular environment [200]. Morphologically similar vesicles and spherically-expanding matrix packets have been seen by TEM within the vulF cells of the vulva [48]. The molecular nature of these vesicles is not known, but their cargo may include the ZP protein LET-653, which is capable of binding spherical aggregates in vitro [201].

In the mammalian lung, lamellar bodies process and deliver lipid-rich surfactant to alveolar air sacs [2]. *C. elegans* external epithelia likewise contain elaborate lamellar structures at their apical membranes [202,203,204]. RAL-1, a GTPase required for exosome secretion, and VHA-5, a component of the V-ATPase, associate with these stacks near the apical surface of seam cells and are required for alae formation [202,204]. These membrane stacks may therefore act as sites of secretory particle organization and/or biogenesis. Both apical membrane stacks and multi-vesicular bodies (MVBs) have been suggested to deliver hedgehog-like proteins and other cuticle components to the cuticle [202,204].

### 7.2. How Are aECMs Anchored to Cell Surfaces?

Many aECM components appear membrane-associated despite lacking obvious domains for membrane spanning or attachment, and in at least some cases, aECM-dependent tissue shaping involves effects on the cytoskeleton. Basal ECMs are generally thought to attach to cell surfaces and the cytoskeleton via integrins, whose extracellular face can bind ECM proteins and intracellular domains bind cytoskeleton modifiers [205]. In contrast, *C. elegans* body cuticles attach to the epidermis and hemidesmosomes via the transmembrane proteins MUP-4 [55] and MUA-3 [55]. It is unclear whether these or other unknown transmembrane proteins anchor other aECMs, such as the embryonic sheath or the pharyngeal cuticle, to apical membranes.

### 7.3. How Are aECMs Assembled and Disassembled?

*C. elegans* aECMs are highly dynamic and spatially specific, with many pre-cuticular aECM components present in restricted regions for mere hours before being replaced by cuticle [48,52]. Furthermore, aECMs can have dazzling complexity, with multiple layers composed of different aECM components [24,48,101,175]. The rapidity of development implies careful regulation of aECM assembly and disassembly by cell-type specific aECM anchors and proteases. However, how this occurs is almost entirely unknown.

aECMs also can be very large. For example, the mid-L4 vulva lumen expands to roughly 10 microns in diameter, with centrally located core aECM components located microns away from their originating cells [20,48]. Elaborate vesicle systems and molecular motors transport proteins to appropriate locations within cells [206], but it is not clear what mechanisms ensure proper placement of aECM proteins within a large extracellular compartment. Most models for aECM layer formation posit sequential rounds of local deposition and detachment [19,31,175], but other biophysical sorting mechanisms or luminal flows may facilitate the movement of some aECM proteins and lipids over longer distances.

### 7.4. What Is the Contribution of Individual aECM Components in Shaping Cell Surfaces?

Although mutant phenotypes for many individual aECM components have been described, the mechanisms by which those components shape their underlying cells often remain unclear. aECMs may shape cells directly by pushing or pulling on apical membranes or creating a stiff scaffold, or indirectly by modulating signaling or interacting with the cytoskeleton across the apical membrane. Identifying interactions between specific aECM proteins and understanding how they anchor to the apical membrane may shed light on how aECM components shape cells.

Many *C. elegans* aECM proteins are related to mammalian matrix proteins and therefore serve as suitable models for studying those specific protein families. For example, ZP proteins (including LET-653, NOAH-1, FBN-1 and CUT-1-6) are abundant in the *C. elegans* pre-cuticle or cuticle [14,20,47,50,91,113,128], and ZP proteins also are present within the mammalian egg coat and in or near aECMs of the gut, vascular and renal systems [207,208,209,210,211,212]. FBN-1 also is related to mammalian fibrillin, a component of mammalian stromal ECMs [213]. The *C. elegans* eLRRon family of pre-cuticle proteins (including LET-4, EGG-6 and SYM-1) [21,47] is related to the small leucine-rich proteoglycans (SLRPs) found in many mammalian ECMs [214]. Lipocalins are a family of known lipid transporters present in or near both *C. elegans* pre-cuticle and mammalian aECMs [53,62,63,215]. Finally, most *C. elegans* cuticle collagens are related to mammalian FACIT collagens. Further work on these proteins in the worm promises to shed light on the trafficking, assembly, and tissue-shaping properties of these conserved matrix protein families.

## Figures and Tables

**Figure 1 jdb-08-00023-f001:**
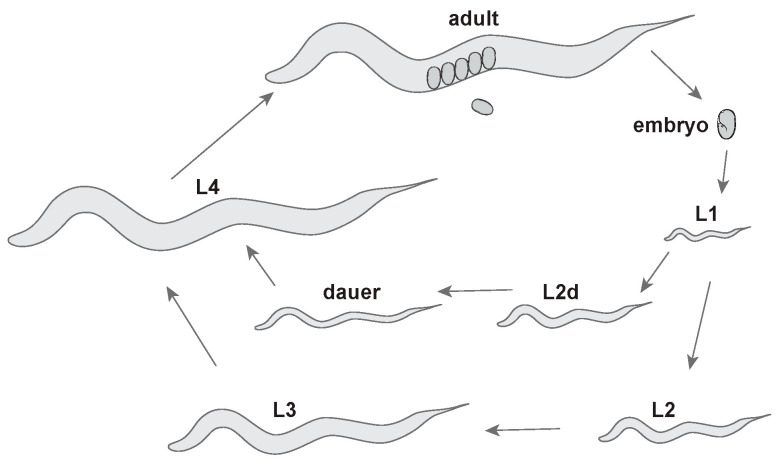
*C. elegans* life cycle. Adults lay embryos that hatch into L1 larvae. Larvae molt into subsequent stages. Under stress (low food, high temperatures, and crowding), larvae can molt into an alternative L3 stage called dauer, which can resume reproductive development upon return to non-stressful conditions. After Wormatlas [28].

**Figure 2 jdb-08-00023-f002:**
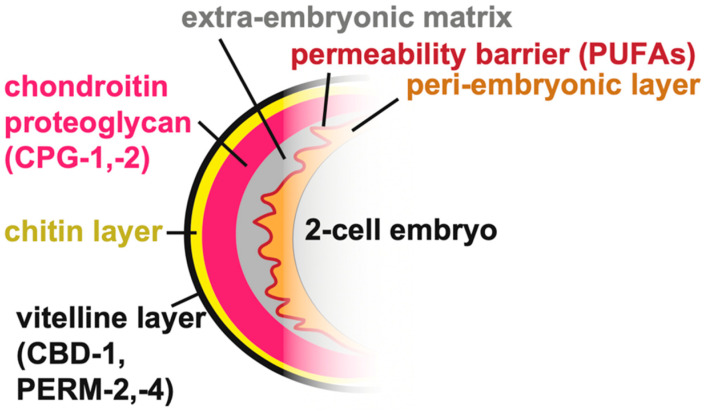
*C. elegans* eggshell. The outermost vitelline layer (black) contains the CBD-1/PERM-2/PERM-4 complex [31]. Next, the chitin-rich layer (yellow) is followed by the chondroitin proteoglycan layer (pink), which contains the chondroitin-proteoglycan proteins CPG-1 and CPG-2 [23]. The extra-embryonic matrix (gray) and the peri-embryonic matrix (orange), which line the embryo, are separated by the permeability barrier (red) [24,31].

**Figure 3 jdb-08-00023-f003:**
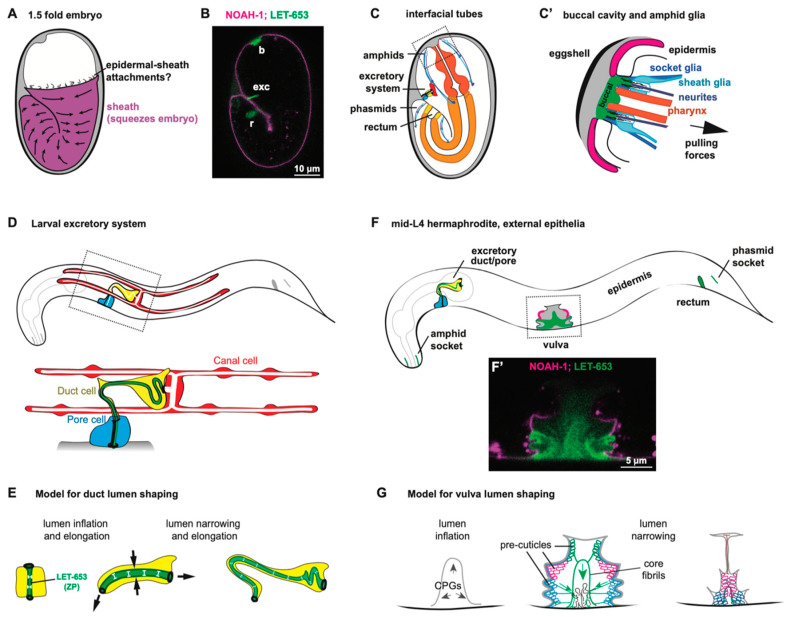
*C. elegans* pre-cuticles shape developing epithelia. (**A**) Diagram of a 1.5 fold *C. elegans* embryo encased in the embryonic sheath. The sheath distributes actinomyosin-based forces that squeeze the embryo into a worm-shape [47]. (**B**) Confocal image of fluorescently-tagged ZP proteins LET-653 and NOAH-1 in a 1.5 fold embryo. LET-653 (green) primarily lines interfacial tubes, including the excretory duct and pore lumen, the rectum (r), and the buccal cavity (b) [48]. NOAH-1 (magenta) is present in the embryonic sheath [47]. (**C**) Cross section of a 1.5-fold embryo, showing interfacial tubes. (**C’**) The pharynx, glia, and neurites are pulled posteriorly while being anchored anteriorly by the pre-cuticle aECM [49,50,51]. Magenta represents NOAH-1-containing pre-cuticle, and green represents LET-653-containing pre-cuticle, as shown in in panel B. (**D**) The larval excretory system. The duct and pore tubes are lined by pre-cuticle and cuticle (green), while the canal tube contains a non-cuticular aECM. Black denotes junctions. (**E**) Model for duct lumen shaping by LET-653 and the pre-cuticle (adapted from [20]). LET-653 (green) promotes duct lumen inflation and resists morphogenetic stretching and squeezing forces (arrows) to maintain proper lumen diameter. (**F**) Diagram of a mid-L4 larva, showing tissues lined by pre-cuticle and cuticle. (**F’**) Confocal image of fluorescently-tagged ZP proteins LET-653 and NOAH-1 in the L4 vulva. (**G**) Model for vulva lumen shaping by the pre-cuticle (adapted from [48]). After initial lumen inflation by CPGs, distinct pre-cuticles form along the apical surfaces of different vulva cell types. Connections between these pre-cuticles and a central core structure contribute to lumen narrowing.

**Figure 4 jdb-08-00023-f004:**
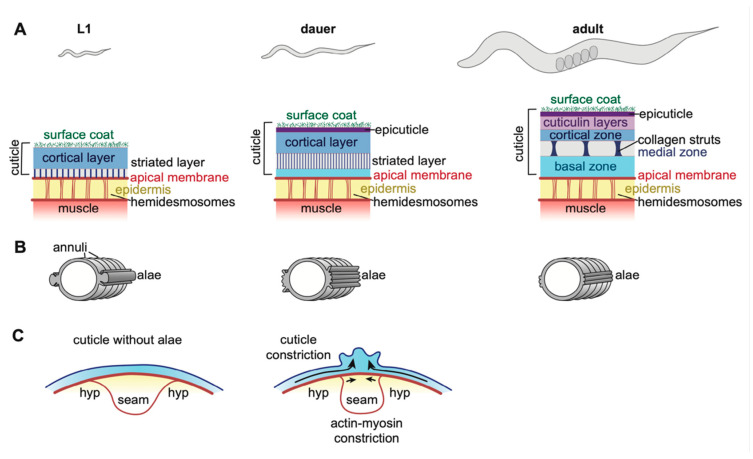
*C. elegans* epidermal cuticle. (**A**) Diagram of *C. elegans* at L1, dauer, and adult stages with cuticle structure at each stage (adapted from [106]). The epidermis connects the muscle and cuticle via hemidesmosomes. At the apical surface, the transmembrane proteins MUP-4 and MUA-3 link hemidesmosomes to the cuticle [55,57]. The cuticle is a multi-layered structure of collagens, cuticulins, lipids, and glycans [91,102,107]. The latter three are likely concentrated near the external surface of the cuticle, while collagens predominate in the basal zone and striated layers. Pre-cuticle and nascent cuticle may appear near the apical membrane prior to molts [53]. (**B**) Cross section of *C. elegans* at each stage indicating the position of alae and annuli. Furrows are the low points between annuli. Alae are not shown to scale. L1 larvae have one large alae ridge flanked by two smaller ones, adults have three alae ridges, while dauer larvae have five. (**C**) Model for alae formation. Constriction by actin-myosin in seam cells and by ZP proteins in the cuticle bend the cuticle into alae ridges [14,108].

**Figure 5 jdb-08-00023-f005:**
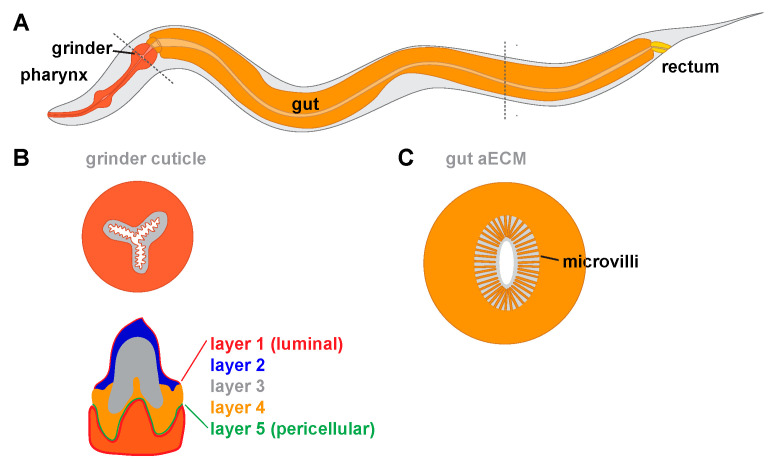
Tubes and aECMs of the *C. elegans* digestive tract. (**A**) Diagram of the digestive system. Different aECMs line the pharynx, gut and rectum. (**B**) Diagram of the pharyngeal grinder containing multiple teeth and the five observed aECM layers within a single tooth (adapted from [175]). Dark orange denotes pharyngeal cell cytoplasm. (**C**) Cross-section through the gut, showing apical microvilli surrounded by a membrane-proximal aECM (adapted from [176]). Light orange denotes intestinal cell cytoplasm.

**Figure 6 jdb-08-00023-f006:**
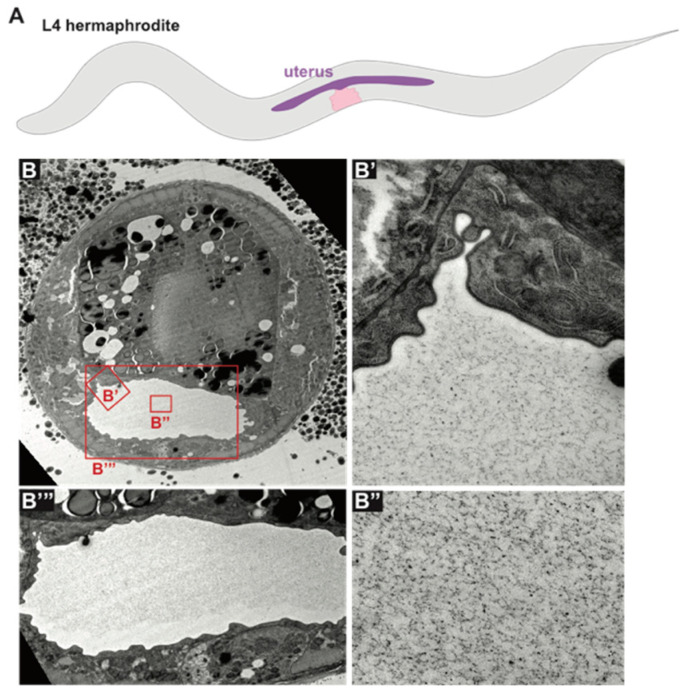
Uterine aECM. (**A**) Diagram of L4 stage *C. elegans* with vulva (pink) and expanded uterus (purple). (**B**) TEM of the uterus at mid-L4 stage. (**B’**–**B’”**) The inflated uterine lumen is filled with a granular matrix of unknown composition. (Electron micrographs courtesy of Alessandro Sparacio).

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
