# Peer review of "C. elegans Apical Extracellular Matrices Shape Epithelia"

_jdb, 2020, doi:10.3390/jdb8040023_

Round 1
Reviewer 1 Report
1. This article is well written and contains detailed descriptions of aECM in different tissues and stages in C. elegans. It successfully gives readers a broad introduction to the C. elegans aECM. However, to help readers get a better understanding of the existing research on aECM, the authors should address more on the lasted progress in this field. More than half of the references are articles before 2010.
2. In section 7, the authors have point out that many worm aECM proteins are similar to mammalian matrix proteins. Moreover, the authors have suggested that aECM studies in worms might be helpful to understand the conserved properties in matrix proteins. Comparing to the cell culture system, what are the advantages of the C. elegans system in extracellular matrix research (fluorescently-tagged protein could be done in cells, too)? How could C. elegans model fill the gaps in current ECM research?
3. Missing Section 6 (page 13 Sec.5 Chitin-base pharyngeal cuticle, page 15 Sec.7 aECMs of internal epithelia)
4. The subheads in Section 8 are not numbered.
Author Response
- This article is well written and contains detailed descriptions of aECM in different tissues and stages in C. elegans. It successfully gives readers a broad introduction to the C. elegans aECM. However, to help readers get a better understanding of the existing research on aECM, the authors should address more on the lasted progress in this field. More than half of the references are articles before 2010.
ïƒ Thank you. We tried to cite all recent articles we could find, along with many older ones. We've now added discussion of the 6 recent papers below. If there are other recent papers that we missed, and that the Reviewer specifically has in mind, please let us know.
The Ancestral Caenorhabditis elegans Cuticle Suppresses rol-1.
Noble LM, Miah A, Kaur T, Rockman MV. G3 (Bethesda). 2020 Jul 7;10(7):2385-2395. doi: 10.1534/g3.120.401336. PMID: 32423919
Now cited at the beginning of section 4.3 (line 710)
Neuronal GPCR NPR-8 regulates C. elegans defense against pathogen infection.
Sellegounder D, Liu Y, Wibisono P, Chen CH, Leap D, Sun J. Sci Adv. 2019 Nov 20;5(11):eaaw4717. doi: 10.1126/sciadv.aaw4717. eCollection 2019 Nov. PMID: 31799388
Now cited in section 4.6 (line 820)
The collagen-derived compound collagen tripeptide induces collagen expression and extends lifespan via a conserved p38 mitogen-activated protein kinase cascade.
Morikiri Y, Matsuta E, Inoue H. Biochem Biophys Res Commun. 2018 Nov 10;505(4):1168-1173. doi: 10.1016/j.bbrc.2018.10.044. Epub 2018 Oct 12. PMID: 30322618
Now cited in section 4.6 (line 820)
Cuticle Collagen Expression Is Regulated in Response to Environmental Stimuli by the GATA Transcription Factor ELT-3 in Caenorhabditis elegans.
Mesbahi H, Pho KB, Tench AJ, Leon Guerrero VL, MacNeil LT. Genetics. 2020 Jun;215(2):483-495. doi: 10.1534/genetics.120.303125. Epub 2020 Mar 30. PMID: 32229533
Now cited in section 4.6 (line 820)
f57f4.4p::gfp as a fluorescent reporter for analysis of the C. elegans response to bacterial infection.
Julien-Gau I, Schmidt M, Kurz CL. Dev Comp Immunol. 2014 Feb;42(2):132-7. doi: 10.1016/j.dci.2013.08.024. Epub 2013 Sep 5. PMID: 24012871
Now cited in section 6.1 (line 908)
Broadly conserved roles of TMEM131 family proteins in intracellular collagen assembly and secretory cargo trafficking.
Zhang Z, Bai M, Barbosa GO, Chen A, Wei Y, Luo S, Wang X, Wang B, Tsukui T, Li H, Sheppard D, Kornberg TB, Ma DK. Sci Adv. 2020 Feb 12;6(7):eaay7667. doi: 10.1126/sciadv.aay7667. eCollection 2020 Feb. PMID: 32095531 Free PMC article. Top of Form
Now cited in section 7.1 (line 962)
- In section 7, the authors have point out that many worm aECM proteins are similar to mammalian matrix proteins. Moreover, the authors have suggested that aECM studies in worms might be helpful to understand the conserved properties in matrix proteins. Comparing to the cell culture system, what are the advantages of the C. elegans system in extracellular matrix research (fluorescently-tagged protein could be done in cells, too)? How could C. elegans model fill the gaps in current ECM research?
ïƒ We changed the text to read, “aECMs are challenging to study as they generally do not develop fully in cell culture and can be destroyed by the fixation required to visualize aECMs in many animal systems. C. elegans is an excellent model for addressing how aECMs function, as it offers a set of aECMs that can be visualized without fixation in vivo.” (lines 940-944)
Note that the introductory sections 1.1 and 1.2 also address this topic.
- Missing Section 6 (page 13 Sec.5 Chitin-base pharyngeal cuticle, page 15 Sec.7 aECMs of internal epithelia)
ïƒ Fixed missing or inaccurate heading numbers
- The subheads in Section 8 are not numbered.
ïƒ Fixed missing or inaccurate heading numbers
Reviewer 2 Report
This is a very exhaustive and well written review on what is known about apical extra cellular matrix in the nematode C. elegans. It will be very useful to the community working in the field, including researchers using mammalian systems. It convincingly presents C. elegans is a good model system to identify new components of the matrix and to study their function. Finally, the authors nicely highlight the important questions that are still open and suggest ways to find answers using C. elegans.
Author Response
This is a very exhaustive and well written review on what is known about apical extra cellular matrix in the nematode C. elegans. It will be very useful to the community working in the field, including researchers using mammalian systems. It convincingly presents C. elegans is a good model system to identify new components of the matrix and to study their function. Finally, the authors nicely highlight the important questions that are still open and suggest ways to find answers using C. elegans.
ïƒ Thank you.
Reviewer 3 Report
Cohen and Sundaram present an excellent review of apical extracellular matrix structure, assembly and trafficking in C. elegans. This review covers a vast amount of research and summarizes it well. I thoroughly enjoyed reading it. I know of no recent reviews on this topic and a clear review of the field was needed.
The review is well written, clear, and logically organized. I have only minor suggestions (outlined below).
It should be mentioned somewhere that C. elegans cuticular collagens differ somewhat from human collagens.
Figure 2 – the vitelline layer is difficult to see in the diagram because all shapes have a colored outline – it gives the impression that the black line is simply the outline of the chitin layer. The outline of the chondroitin layer could also be mistaken as a black line.
Figure 3 – there are two panels labelled D
Line 303 – che-14 is mentioned along with daf-6, but only the daf-6 phenotype is discussed.
line 383-4 – The authors mention that the cuticle is unique for each life stage. Could they clarify in the text how they are defined as “different” – i.e. are they referring to observations of structure or content? Collagen content differs but is there additional data related to stage-specific protein and lipid content?
Figure 4 – Do the diagrams in A need to be a separate panel? Could they not simply be part of B?
Some clarification about the alae ridges in adult vs L1 could be added. The text states that there are three ridges at each of these stage but not how they are structurally different. Further, the L1 alae ridges in the diagram in Figure 4C look like a single large ridge.
Figure 4 - The limited contrast between the grey color of the epicuticle and the adjacent blue makes this layer a little hard to see.
Line 417 – “In adults, collagens predominate in the basal-most striated layers” – link this to the image in Figure 4B by either modifying the text or the diagram to clearly indicate which layers are considered the basal-most striated layers
line 537 – In “with distinct phases of the molt cycles” I think cycles should be singular, since “cycle” refers to a re-iterative process, but I will leave this to the authors to decide.
Line 542 – the authors use the term “during molt” – could they clarify whether they mean during ecdysis itself or whether they are referring to the period during development when animals are progressing through the molting cycle.
Section 4.7 would benefit from a more descriptive title.
Information is repeated in the first paragraph of 3.2 and in 7.3 - although this is a minor criticism as it may serve simply to remind the reader of the role of excretory canal.
Line 611 – The final sentence in the introduction to section 7 could be made more clear.
Author Response
Cohen and Sundaram present an excellent review of apical extracellular matrix structure, assembly and trafficking in C. elegans. This review covers a vast amount of research and summarizes it well. I thoroughly enjoyed reading it. I know of no recent reviews on this topic and a clear review of the field was needed.
The review is well written, clear, and logically organized. I have only minor suggestions (outlined below).
ïƒ Thank you.
It should be mentioned somewhere that C. elegans cuticular collagens differ somewhat from human collagens.
ïƒ We changed the text to read, “Most of the collagens are related to the mammalian FACIT (Fibril-Associated Collagens with Interrupted Triple helices) family, although some have unusual features not seen in mammalian collagens.” (lines 625-626)
Figure 2 – the vitelline layer is difficult to see in the diagram because all shapes have a colored outline – it gives the impression that the black line is simply the outline of the chitin layer. The outline of the chondroitin layer could also be mistaken as a black line.
ïƒ We changed the coloring of this cartoon and eliminated the outline of the chondroitin layer.
Figure 3 – there are two panels labelled D
ïƒ We removed the additional letter “D”.
Line 303 – che-14 is mentioned along with daf-6, but only the daf-6 phenotype is discussed.
ïƒ We clarified this point, “daf-6 mutants have closed sockets and expanded sheath lumens (Oikonomou et al., 2011), while che-14 mutants accumulate vesicles in the amphid lumen (Michaux et al., 2000). In addition, daf-6 mutants also have dyf-7-like dendrite anchoring defects (Hong et al., 2020).” (lines 373-375)
line 383-4 – The authors mention that the cuticle is unique for each life stage. Could they clarify in the text how they are defined as “different” – i.e. are they referring to observations of structure or content? Collagen content differs but is there additional data related to stage-specific protein and lipid content?
ïƒ We changed the text to read, “Between each larval stage, C. elegans molts into a new cuticle that is unique in structure and collagen composition for that life stage, but how these cuticles differ functionally is not clear.” (lines 588-589)
Figure 4 – Do the diagrams in A need to be a separate panel? Could they not simply be part of B?
ïƒ The diagrams in A were combined with panel B as suggested.
Some clarification about the alae ridges in adult vs L1 could be added. The text states that there are three ridges at each of these stage but not how they are structurally different. Further, the L1 alae ridges in the diagram in Figure 4C look like a single large ridge.
We changed the Figure 4 legend to clarify L1 alae structure: “L1 alae have one large alae ridge flanked by two smaller ones." (lines 616-617)
We also added more details to the main text:
“For example, CUT-1 promotes formation of dauer alae, CUT-3 promotes formation of L1 alae, and CUT-4 promotes formation of adult alae (Sapio et al., 2005). Both CUT-5 and the nidogen domain protein DEX-1 promote formation of alae in L1s and dauers, but not adults (Flatt et al., 2019; Sapio et al., 2005). The collagens DPY-2, DPY-3, DPY-10 (McMahon et al., 2003), DPY-5, DPY-11 and DPY-13 (Dodd et al., 2018), and the secreted proline-rich-repeat protein MLT-10 (Meli et al., 2010) are all required for normal adult alae morphology; it’s not clear whether they are also required for development of L1 or dauer alae.” (lines 694-701)
Figure 4 - The limited contrast between the grey color of the epicuticle and the adjacent blue makes this layer a little hard to see.
ïƒ The epicuticle was changed from gray to purple.
Line 417 – “In adults, collagens predominate in the basal-most striated layers” – link this to the image in Figure 4B by either modifying the text or the diagram to clearly indicate which layers are considered the basal-most striated layers
ïƒ The Figure 4 legend was modified to read, “The latter three are likely concentrated near the external surface of the cuticle, while collagens predominate in the basal zone and striated layers.” (lines 612-613)
line 537 – In “with distinct phases of the molt cycles” I think cycles should be singular, since “cycle” refers to a re-iterative process, but I will leave this to the authors to decide.
ïƒ We changed the text as suggested (line 592).
Line 542 – the authors use the term “during molt” – could they clarify whether they mean during ecdysis itself or whether they are referring to the period during development when animals are progressing through the molting cycle.
ïƒ We changed the text to, “Lysosome-related organelle (LRO) morphology changes dramatically in epidermal cells before and during ecdysis, indicating that LROs may be particularly important for molting.” (lines 823-825)
Section 4.7 would benefit from a more descriptive title.
ïƒ We changed the title to, “4.3 Epidermal cuticles maintain body length and girth” (line 709)
Information is repeated in the first paragraph of 3.2 and in 7.3 - although this is a minor criticism as it may serve simply to remind the reader of the role of excretory canal.
ïƒ We feel that readers unfamiliar with C. elegans may require reminders, since the place where the canal is first introduced (section 3.2) is far from where it is later discussed (6.3).
Line 611 – The final sentence in the introduction to section 7 could be made more clear.
ïƒ We changed the text to read, “C. elegans is an excellent model for addressing how aECMs assemble and function, as it offers a set of aECMs that can be visualized without fixation in vivo.” (lines 944-945)